# Enhancing the Locality and Breaking the Memory Bottleneck of Transformer on Time Series Forecasting

**Shiyang Li**
shiyangli@ucsb.edu

**Xiaoyong Jin**
x_jin@ucsb.edu

**Yao Xuan**
yxuan@ucsb.edu

**Xiyou Zhou**
xiyou@ucsb.edu

**Wenhu Chen**
wenhuchen@ucsb.edu

**Yu-Xiang Wang**
yuxiangw@cs.ucsb.edu

**Xifeng Yan**
xyan@cs.ucsb.edu

University of California, Santa Barbara

## Abstract

Time series forecasting is an important problem across many domains, including predictions of solar plant energy output, electricity consumption, and traffic jam situation. In this paper, we propose to tackle such forecasting problem with *Transformer* [1]. Although impressed by its performance in our preliminary study, we found its two major weaknesses: (1) *locality-agnostics*: the point-wise dot-product self-attention in canonical Transformer architecture is insensitive to local context, which can make the model prone to anomalies in time series; (2) *memory bottleneck*: space complexity of canonical Transformer grows quadratically with sequence length $L$, making directly modeling long time series infeasible. In order to solve these two issues, we first propose convolutional self-attention by producing queries and keys with *causal convolution* so that local context can be better incorporated into attention mechanism. Then, we propose *LogSparse* Transformer with only $O(L(\log L)^2)$ memory cost, improving forecasting accuracy for time series with fine granularity and strong long-term dependencies under constrained memory budget. Our experiments on both synthetic data and real-world datasets show that it compares favorably to the state-of-the-art.

## 1 Introduction

Time series forecasting plays an important role in daily life to help people manage resources and make decisions. For example, in retail industry, probabilistic forecasting of product demand and supply based on historical data can help people do inventory planning to maximize the profit. Although still widely used, traditional time series forecasting models, such as State Space Models (SSMs) [2] and Autoregressive (AR) models, are designed to fit each time series independently. Besides, they also require practitioners' expertise in manually selecting trend, seasonality and other components. To sum up, these two major weaknesses have greatly hindered their applications in the modern large-scale time series forecasting tasks.

To tackle the aforementioned challenges, deep neural networks [3, 4, 5, 6] have been proposed as an alternative solution, where Recurrent Neural Network (RNN) [7, 8, 9] has been employed to model time series in an autoregressive fashion. However, RNNs are notoriously difficult to train [10] because of gradient vanishing and exploding problem. Despite the emergence of various variants, including LSTM [11] and GRU [12], the issues still remain unresolved. As an example, [13] shows that language models using LSTM have an effective context size of about 200 tokens on average but are only able to sharply distinguish 50 tokens nearby, indicating that even LSTM struggles to

capture long-term dependencies. On the other hand, real-world forecasting applications often have both long- and short-term repeating patterns [7]. For example, the hourly occupancy rate of a freeway in traffic data has both daily and hourly patterns. In such cases, how to model long-term dependencies becomes the critical step in achieving promising performances.

Recently, Transformer [1, 14] has been proposed as a brand new architecture which leverages attention mechanism to process a sequence of data. Unlike the RNN-based methods, Transformer allows the model to access any part of the history regardless of distance, making it potentially more suitable for grasping the recurring patterns with long-term dependencies. However, canonical dot-product self-attention matches queries against keys insensitive to local context, which may make the model prone to anomalies and bring underlying optimization issues. More importantly, space complexity of canonical Transformer grows quadratically with the input length $L$, which causes memory bottleneck on directly modeling long time series with fine granularity. We specifically delve into these two issues and investigate the applications of Transformer to time series forecasting. Our contributions are three fold:

- We successfully apply Transformer architecture to time series forecasting and perform extensive experiments on both synthetic and real datasets to validate Transformer's potential value in better handling long-term dependencies than RNN-based models.

- We propose *convolutional* self-attention by employing causal convolutions to produce queries and keys in the self-attention layer. Query-key matching aware of local context, e.g. shapes, can help the model achieve lower training loss and further improve its forecasting accuracy.

- We propose *LogSparse* Transformer, with only $O(L(\log L)^2)$ space complexity to break the memory bottleneck, not only making fine-grained long time series modeling feasible but also producing comparable or even better results with much less memory usage, compared to canonical Transformer.

## 2 Related Work

Due to the wide applications of forecasting, various methods have been proposed to solve the problem. One of the most prominent models is `ARIMA` [15]. Its statistical properties as well as the well-known Box-Jenkins methodology [16] in the model selection procedure make it the first attempt for practitioners. However, its linear assumption and limited scalability make it unsuitable for large-scale forecasting tasks. Further, information across similar time series cannot be shared since each time series is fitted individually. In contrast, [17] models related time series data as a matrix and deal with forecasting as a matrix factorization problem. [18] proposes hierarchical Bayesian methods to learn across multiple related count time series from the perspective of graph model.

Deep neural networks have been proposed to capture shared information across related time series for accurate forecasting. [3] fuses traditional AR models with RNNs by modeling a probabilistic distribution in an encoder-decoder fashion. Instead, [19] uses an RNN as an encoder and Multi-layer Perceptrons (MLPs) as a decoder to solve the so-called error accumulation issue and conduct multi-ahead forecasting in parallel. [6] uses a global RNN to directly output the parameters of a linear SSM at each step for each time series, aiming to approximate nonlinear dynamics with locally linear segments. In contrast, [9] deals with noise using a local Gaussian process for each time series while using a global RNN to model the shared patterns. [20] tries to combine the advantages of AR models and SSMs, and maintain a complex latent process to conduct multi-step forecasting in parallel.

The well-known self-attention based Transformer [1] has recently been proposed for sequence modeling and has achieved great success. Several recent works apply it to translation, speech, music and image generation [1, 21, 22, 23]. However, scaling attention to extremely long sequences is computationally prohibitive since the space complexity of self-attention grows quadratically with sequence length [21]. This becomes a serious issue in forecasting time series with fine granularity and strong long-term dependencies.

## 3 Background

**Problem definition**    Suppose we have a collection of $N$ related univariate time series $\{\mathbf{z}_{i,1:t_0}\}_{i=1}^N$, where $\mathbf{z}_{i,1:t_0} \triangleq [\mathbf{z}_{i,1}, \mathbf{z}_{i,2}, \cdots, \mathbf{z}_{i,t_0}]$ and $\mathbf{z}_{i,t} \in \mathbb{R}$ denotes the value of time series $i$ at time $t$[1]. We are going to predict the next $\tau$ time steps for all time series, i.e. $\{\mathbf{z}_{i,t_0+1:t_0+\tau}\}_{i=1}^N$. Besides, let $\{\mathbf{x}_{i,1:t_0+\tau}\}_{i=1}^N$ be a set of associated time-based covariate vectors with dimension $d$ that are assumed to be known over the entire time period, e.g. day-of-the-week and hour-of-the-day. We aim to model the following conditional distribution

$$p(\mathbf{z}_{i,t_0+1:t_0+\tau}|\mathbf{z}_{i,1:t_0}, \mathbf{x}_{i,1:t_0+\tau}; \mathbf{\Phi}) = \prod_{t=t_0+1}^{t_0+\tau} p(\mathbf{z}_{i,t}|\mathbf{z}_{i,1:t-1}, \mathbf{x}_{i,1:t}; \mathbf{\Phi}).$$

We reduce the problem to learning a one-step-ahead prediction model $p(\mathbf{z}_t|\mathbf{z}_{1:t-1}, \mathbf{x}_{1:t}; \mathbf{\Phi})$ [2], where $\mathbf{\Phi}$ denotes the learnable parameters shared by all time series in the collection. To fully utilize both the observations and covariates, we concatenate them to obtain an augmented matrix as follows:

$$\mathbf{y}_t \triangleq [\mathbf{z}_{t-1} \circ \mathbf{x}_t] \in \mathbb{R}^{d+1}, \qquad \mathbf{Y}_t = [\mathbf{y}_1, \cdots, \mathbf{y}_t]^T \in \mathbb{R}^{t \times (d+1)},$$

where $[\cdot \circ \cdot]$ represents concatenation. An appropriate model $\mathbf{z}_t \sim f(\mathbf{Y}_t)$ is then explored to predict the distribution of $\mathbf{z}_t$ given $\mathbf{Y}_t$.

**Transformer**    We instantiate $f$ with Transformer [3] by taking advantage of the multi-head self-attention mechanism, since self-attention enables Transformer to capture both long- and short-term dependencies, and different attention heads learn to focus on different aspects of temporal patterns. These advantages make Transformer a good candidate for time series forecasting. We briefly introduce its architecture here and refer readers to [1] for more details.

In the self-attention layer, a multi-head self-attention sublayer simultaneously transforms $\mathbf{Y}$ [4] into $H$ distinct query matrices $\mathbf{Q}_h = \mathbf{Y}\mathbf{W}_h^Q$, key matrices $\mathbf{K}_h = \mathbf{Y}\mathbf{W}_h^K$, and value matrices $\mathbf{V}_h = \mathbf{Y}\mathbf{W}_h^V$ respectively, with $h = 1, \cdots, H$. Here $\mathbf{W}_h^Q, \mathbf{W}_h^K \in \mathbb{R}^{(d+1) \times d_k}$ and $\mathbf{W}_h^V \in \mathbb{R}^{(d+1) \times d_v}$ are learnable parameters. After these linear projections, the scaled dot-product attention computes a sequence of vector outputs:

$$\mathbf{O}_h = \text{Attention}(\mathbf{Q}_h, \mathbf{K}_h, \mathbf{V}_h) = \text{softmax}\left(\frac{\mathbf{Q}_h\mathbf{K}_h^T}{\sqrt{d_k}} \cdot \mathbf{M}\right)\mathbf{V}_h.$$

Note that a mask matrix $\mathbf{M}$ is applied to filter out rightward attention by setting all upper triangular elements to $-\infty$, in order to avoid future information leakage. Afterwards, $\mathbf{O}_1, \mathbf{O}_2, \cdots, \mathbf{O}_H$ are concatenated and linearly projected again. Upon the attention output, a position-wise feedforward sublayer with two layers of fully-connected network and a ReLU activation in the middle is stacked.

## 4 Methodology

### 4.1 Enhancing the locality of Transformer

Patterns in time series may evolve with time significantly due to various events, e.g. holidays and extreme weather, so whether an observed point is an anomaly, change point or part of the patterns is highly dependent on its surrounding context. However, in the self-attention layers of canonical Transformer, the similarities between queries and keys are computed based on their point-wise values without fully leveraging local context like shape, as shown in Figure 1(a) and (b). Query-key matching agnostic of local context may confuse the self-attention module in terms of whether the observed value is an anomaly, change point or part of patterns, and bring underlying optimization issues.

We propose convolutional self-attention to ease the issue. The architectural view of proposed convolutional self-attention is illustrated in Figure 1(c) and (d). Rather than using convolution of

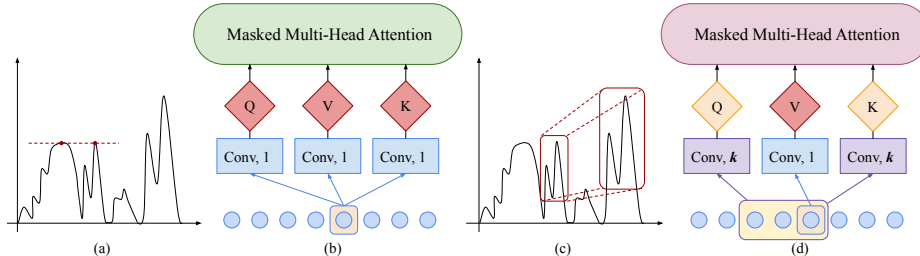

Figure 1: The comparison between canonical and our convolutional self-attention layers. "Conv, 1" and "Conv, $k$" mean convolution of kernel size $\{1, k\}$ with stride 1, respectively. Canonical self-attention as used in Transformer is shown in (b), may wrongly match point-wise inputs as shown in (a). Convolutional self-attention is shown in (d), which uses convolutional layers of kernel size $k$ with stride 1 to transform inputs (with proper paddings) into queries/keys. Such locality awareness can correctly match the most relevant features based on shape matching in (c).

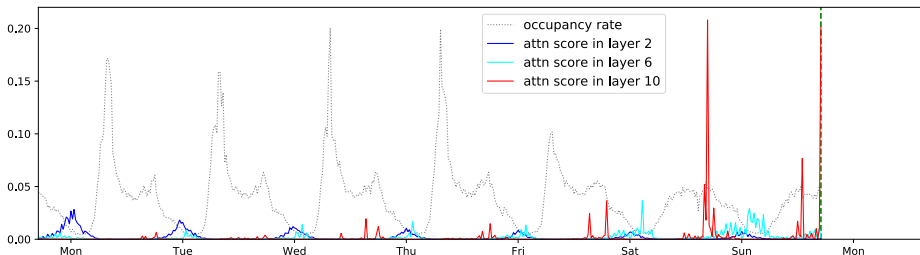

Figure 2: Learned attention patterns from a 10-layer canonical Transformer trained on `traffic-f` dataset with full attention. The green dashed line indicates the start time of forecasting and the gray dashed line on its left side is the conditional history. Blue, cyan and red lines correspond to attention patterns in layer 2, 6 and 10, respectively, for a head when predicting the value at the time corresponding to the green dashed line. a) Layer 2 tends to learn shared patterns in every day. b) Layer 6 focuses more on weekend patterns. c) Layer 10 further squeezes most of its attention on only several cells in weekends, causing most of the others to receive little attention.

kernel size 1 with stride 1 (matrix multiplication), we employ *causal convolution* of kernel size $k$ with stride 1 to transform inputs (with proper paddings) into queries and keys. Note that causal convolutions ensure that the current position never has access to future information. By employing causal convolution, generated queries and keys can be more aware of local context and hence, compute their similarities by their local context information, e.g. local shapes, instead of point-wise values, which can be helpful for accurate forecasting. Note that when $k = 1$, the convolutional self-attention will degrade to canonical self-attention, thus it can be seen as a generalization.

## 4.2 Breaking the memory bottleneck of Transformer

To motivate our approach, we first perform a qualitative assessment of the learned attention patterns with a canonical Transformer on `traffic-f` dataset. The `traffic-f` dataset contains occupancy rates of 963 car lanes of San Francisco bay area recorded every 20 minutes [6]. We trained a 10-layer canonical Transformer on `traffic-f` dataset with full attention and visualized the learned attention patterns. One example is shown in Figure 2. Layer 2 clearly exhibited global patterns, however, layer 6 and 10, only exhibited pattern-dependent sparsity, suggesting that some form of sparsity could be introduced without significantly affecting performance. More importantly, for a sequence with length $L$, computing attention scores between every pair of cells will cause $O(L^2)$ memory usage, making modeling long time series with fine granularity and strong long-term dependencies prohibitive.

We propose *LogSparse* Transformer, which only needs to calculate $O(\log L)$ dot products for each cell in each layer. Further, we only need to stack up to $O(\log L)$ layers and the model will be able to access every cell's information. Hence, the total cost of memory usage is only $O(L(\log L)^2)$. We define $I_l^k$ as the set of indices of the cells that cell $l$ can attend to during the computation from $k_{th}$

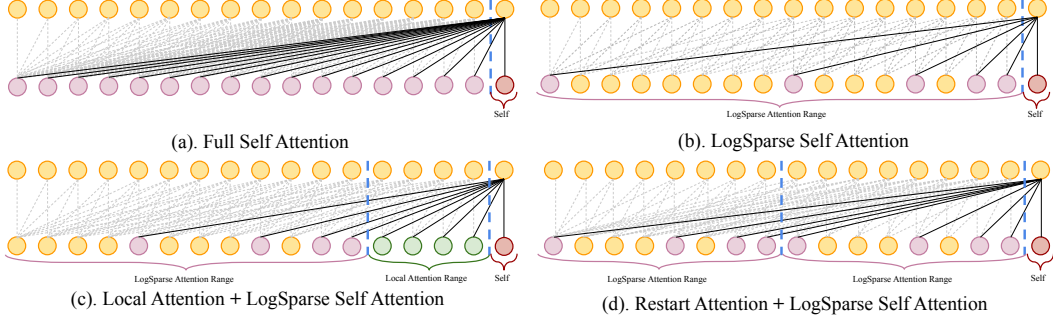

Figure 3: Illustration of different attention mechanism between adjacent layers in Transformer.

layer to $(k+1)_{th}$ layer. In the standard self-attention of Transformer, $I_l^k = \{j : j \le l\}$, allowing every cell to attend to all its past cells and itself as shown in Figure 3(a). However, such an algorithm suffers from the quadratic space complexity growth along with the input length. To alleviate such an issue, we propose to select a subset of the indices $I_l^k \subset \{j : j \le l\}$ so that $|I_l^k|$ does not grow too fast along with $l$. An effective way of choosing indices is $|I_l^k| \propto \log L$.

Notice that cell $l$ is a weighted combination of cells indexed by $I_l^k$ in $k$th self-attention layer and can pass the information of cells indexed by $I_l^k$ to its followings in the next layer. Let $S_l^k$ be the set which contains indices of all the cells whose information has passed to cell $l$ up to $k_{th}$ layer. To ensure that every cell receives the information from all its previous cells and itself, the number of stacked layers $\tilde{k}_l$ should satisfy that $S_l^{\tilde{k}_l} = \{j : j \le l\}$ for $l = 1, \cdots, L$. That is, $\forall l$ and $j \le l$, there is a directed path $P_{jl} = (j, p_1, p_2, \cdots, l)$ with $\tilde{k}_l$ edges, where $j \in I_{p_1}^1, p_1 \in I_{p_2}^2, \cdots, p_{\tilde{k}_l-1} \in I_l^{\tilde{k}_l}$.

We propose *LogSparse* self-attention by allowing each cell only to attend to its previous cells with an exponential step size and itself. That is, $\forall k$ and $l$, $I_l^k = \{l - 2^{\lfloor \log_2 l \rfloor}, l - 2^{\lfloor \log_2 l \rfloor - 1}, l - 2^{\lfloor \log_2 l \rfloor - 2}, ..., l - 2^0, l\}$, where $\lfloor \cdot \rfloor$ denotes the floor operation, as shown in Figure 3(b).[5]

**Theorem 1.** *$\forall l$ and $j \le l$, there is at least one path from cell $j$ to cell $l$ if we stack $\lfloor \log_2 l \rfloor + 1$ layers. Moreover, for $j < l$, the number of feasible unique paths from cell $j$ to cell $l$ increases at a rate of $O(\lfloor \log_2(l-j) \rfloor!)$.*

The proof, deferred to Appendix A.1, uses a constructive argument.

Theorem 1 implies that despite an exponential decrease in the memory usage (from $O(L^2)$ to $O(L \log_2 L)$) in each layer, the information could still flow from any cell to any other cell provided that we go slightly "deeper" — take the number of layers to be $\lfloor \log_2 L \rfloor + 1$. Note that this implies an overall memory usage of $O(L(\log_2 L)^2)$ and addresses the notorious scalability bottleneck of Transformer under GPU memory constraint [1]. Moreover, as two cells become further apart, the number of paths increases at a rate of super-exponential in $\log_2(l-j)$, which indicates a rich information flow for modeling delicate long-term dependencies.

**Local Attention** We can allow each cell to densely attend to cells in its left window of size $O(\log_2 L)$ so that more local information, e.g. trend, can be leveraged for current step forecasting. Beyond the neighbor cells, we can resume our *LogSparse* attention strategy as shown in Figure 3(c).

**Restart Attention** Further, one can divide the whole input with length $L$ into subsequences and set each subsequence length $L_{sub} \propto L$. For each of them, we apply the *LogSparse* attention strategy. One example is shown in Figure 3(d).

Employing *local attention* and *restart attention* won't change the complexity of our sparse attention strategy but will create more paths and decrease the required number of edges in the path. Note that one can combine local attention and restart attention together.

## 5 Experiments

### 5.1 Synthetic datasets

To demonstrate Transformer's capability to capture long-term dependencies, we conduct experiments on synthetic data. Specifically, we generate a piece-wise sinusoidal signals

$$
f(x) = \begin{cases}
A_1 \sin(\pi x/6) + 72 + N_x & x \in [0, 12), \\
A_2 \sin(\pi x/6) + 72 + N_x & x \in [12, 24), \\
A_3 \sin(\pi x/6) + 72 + N_x & x \in [24, t_0), \\
A_4 \sin(\pi x/12) + 72 + N_x & x \in [t_0, t_0 + 24),
\end{cases}
$$

where $x$ is an integer, $A_1, A_2, A_3$ are randomly generated by uniform distribution on $[0, 60]$, $A_4 = \max(A_1, A_2)$ and $N_x \sim \mathcal{N}(0, 1)$. Following the forecasting setting in Section 3, we aim to predict the last 24 steps given the previous $t_0$ data points. Intuitively, larger $t_0$ makes forecasting more difficult since the model is required to understand and remember the relation between $A_1$ and $A_2$ to make correct predictions after $t_0 - 24$ steps of irrelevant signals. Hence, we create 8 different datasets by varying the value of $t_0$ within $\{24, 48, 72, 96, 120, 144, 168, 192\}$. For each dataset, we generate 4.5K, 0.5K and 1K time series instances for training, validation and test set, respectively. An example time series with $t_0 = 96$ is shown in Figure 4(a).

In this experiment, we use a 3-layer canonical Transformer with standard self-attention. For comparison, we employ `DeepAR` [3], an autoregressive model based on a 3-layer LSTM, as our baseline. Besides, to examine if larger capacity could improve performance of `DeepAR`, we also gradually increase its hidden size $h$ as $\{20, 40, 80, 140, 200\}$. Following [3, 6], we evaluate both methods using $\rho$-quantile loss $R_\rho$ with $\rho \in (0, 1)$,

$$
R_\rho(\mathbf{x}, \hat{\mathbf{x}}) = \frac{2 \sum_{i,t} D_\rho(x_t^{(i)}, \hat{x}_t^{(i)})}{\sum_{i,t} |x_t^{(i)}|}, \quad D_\rho(x, \hat{x}) = (\rho - \mathbf{I}_{\{x \le \hat{x}\}})(x - \hat{x}),
$$

where $\hat{x}$ is the empirical $\rho$-quantile of the predictive distribution and $\mathbf{I}_{\{x \le \hat{x}\}}$ is an indicator function.

Figure 4(b) presents the performance of `DeepAR` and Transformer on the synthetic datasets. When $t_0 = 24$, both of them perform very well. But, as $t_0$ increases, especially when $t_0 \ge 96$, the performance of `DeepAR` drops significantly while Transformer keeps its accuracy, suggesting that Transformer can capture fairly long-term dependencies when LSTM fails to do so.

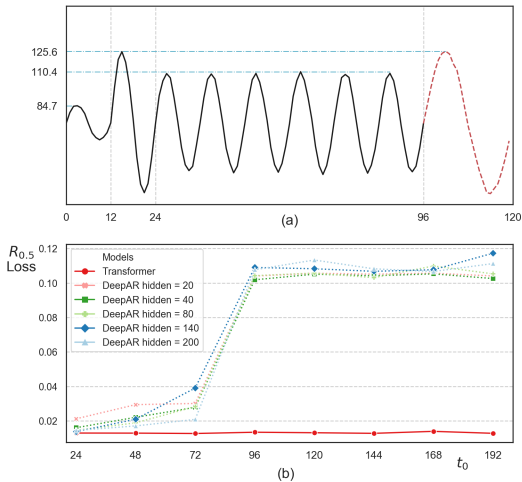

### 5.2 Real-world datasets

We further evaluate our model on several real-world datasets. The `electricity-f (fine)` dataset consists of electricity consumption of 370 customers recorded every 15 minutes and the `electricity-c (coarse)` dataset is the aggregated `electricity-f` by every 4 points, producing hourly electricity consumption. Similarly, the `traffic-f (fine)` dataset contains occupancy rates of 963 freeway in San Francisco recorded every 20 minutes and the `traffic-c (coarse)` contains hourly occupancy rates by averaging every 3 points in `traffic-f`. The `solar` dataset[6] contains the solar power production records from January to August in 2006, which is sampled every hour from 137 PV plants in Alabama. The `wind`[7] dataset contains daily

Figure 4: (a) An example time series with $t_0 = 96$. Black line is the conditional history while red dashed line is the target. (b) Performance comparison between `DeepAR` and canonical Transformer along with the growth of $t_0$. The larger $t_0$ is, the longer dependencies the models need to capture for accurate forecasting.

[6]https://www.nrel.gov/grid/solar-power-data.html
[7]https://www.kaggle.com/sohier/30-years-of-european-wind-generation

Table 1: Results summary ($R_{0.5}/R_{0.9}$-loss) of all methods. e-c and t-c represent `electricity-c` and `traffic-c`, respectively. In the 1st and 3rd row, we perform rolling-day prediction of 7 days while in the 2nd and 4th row, we directly forecast 7 days ahead. TRMF outputs points predictions, so we only report $R_{0.5}$. $\diamond$ denotes results from [6].

|  | ARIMA | ETS | TRMF | DeepAR | DeepState | Ours |
|---|---|---|---|---|---|---|
| e-c$_{1d}$ | 0.154/0.102 | 0.101/0.077 | 0.084/- | 0.075$^\diamond$/0.040$^\diamond$ | 0.083$^\diamond$/0.056$^\diamond$ | **0.059/0.034** |
| e-c$_{7d}$ | 0.283$^\diamond$/0.109$^\diamond$ | 0.121$^\diamond$/0.101$^\diamond$ | 0.087/- | 0.082/0.053 | 0.085$^\diamond$/0.052$^\diamond$ | **0.070/0.044** |
| t-c$_{1d}$ | 0.223/0.137 | 0.236/0.148 | 0.186/- | 0.161$^\diamond$/0.099$^\diamond$ | 0.167$^\diamond$/0.113$^\diamond$ | **0.122/0.081** |
| t-c$_{7d}$ | 0.492$^\diamond$/0.280$^\diamond$ | 0.509$^\diamond$/0.529$^\diamond$ | 0.202/- | 0.179/0.105 | 0.168$^\diamond$/0.114$^\diamond$ | **0.139/0.094** |

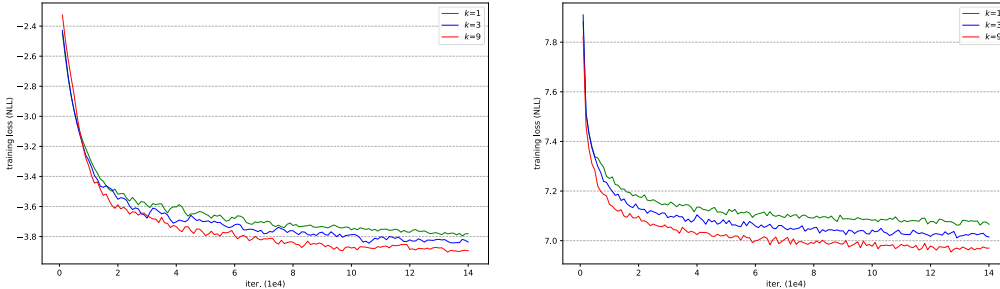

Figure 5: Training curve comparison (with proper smoothing) among kernel size $k \in \{1, 3, 9\}$ in `traffic-c` (**left**) and `electricity-c` (**right**) dataset. Being aware of larger local context size, the model can achieve lower training error and converge faster.

estimates of 28 countries' energy potential from 1986 to 2015 as a percentage of a power plant's maximum output. The `M4-Hourly` contains 414 hourly time series from M4 competition [24].

**Long-term and short-term forecasting**  We first show the effectiveness of canonical Transformer equipped with convolutional self-attention in long-term and short-term forecasting in `electricity-c` and `traffic-c` dataset. These two datasets exhibit both hourly and daily seasonal patterns. However, `traffic-c` demonstrates much greater difference between the patterns of weekdays and weekends compared to `electricity-c`. Hence, accurate forecasting in `traffic-c` dataset requires the model to capture both long- and short-term dependencies very well. As baselines, we use classical forecasting methods `auto.arima`, `ets` implemented in R's `forecast` package and the recent matrix factorization method `TRMF` [17], a RNN-based autoregressive model `DeepAR` and a RNN-based state space model `DeepState` [6]. For short-term forecasting, we evaluate rolling-day forecasts for seven days ( i.e., prediction horizon is one day and forecasts start time is shifted by one day after evaluating the prediction for the current day [6]). For long-term forecasting, we directly forecast 7 days ahead. As shown in Table 1, our models with convolutional self-attention get betters results in both long-term and short-term forecasting, especially in `traffic-c` dataset compared to strong baselines, partly due to the long-term dependency modeling ability of Transformer as shown in our synthetic data.

**Convolutional self-attention**  In this experiment, we conduct ablation study of our proposed convolutional self-attention. We explore different kernel size $k \in \{1, 2, 3, 6, 9\}$ on the full attention model and fix all other settings. We still use rolling-day prediction for seven days on `electricity-c` and `traffic-c` datasets. The results of different kernel sizes on both datasets are shown in Table 2. On `electricity-c` dataset, models with kernel size $k \in \{2, 3, 6, 9\}$ obtain slightly better results in term of $R_{0.5}$ than canonical Transformer but overall these results are comparable and all of them perform very well. We argue it is because `electricity-c` dataset is less challenging and covariate vectors have already provided models with rich information for accurate forecasting. Hence, being aware of larger local context may not help a lot in such cases. However, on much more challenging `traffic-c` dataset, the model with larger kernel size $k$ can make more accurate forecasting than models with smaller ones with as large as $9\%$ relative improvement. These consistent gains can be the results of more accurate query-key matching by being aware of more local context. Further, to verify if incorporating more local context into query-key matching can ease the training, we plot the

Table 2: Average $R_{0.5}/R_{0.9}$-loss of different kernel sizes for rolling-day prediction of 7 days.

| | $k = 1$ | $k = 2$ | $k = 3$ | $k = 6$ | $k = 9$ |
|---|---|---|---|---|---|
| electricity-c$_{1d}$ | 0.060/**0.030** | 0.058/**0.030** | **0.057**/0.031 | **0.057**/0.031 | 0.059/0.034 |
| traffic-c$_{1d}$ | 0.134/0.089 | 0.124/0.085 | 0.123/0.083 | 0.123/0.083 | **0.122/0.081** |

training loss of kernel size $k \in \{1, 3, 9\}$ in `electricity-c` and `traffic-c` datasets. We found that Transformer with convolutional self-attention also converged faster and to lower training errors, as shown in Figure 5, proving that being aware of local context can ease the training process.

**Sparse attention** Further, we compare our proposed *LogSparse* Transformer to the full attention counterpart on fine-grained datasets, `electricity-f` and `traffic-f`. Note that time series in these two datasets have much longer periods and are noisier comparing to `electricity-c` and `traffic-c`. We first compare them under the same memory budget. For `electricity-f` dataset, we choose $L_{e_1} = 768$ with subsequence length $L_{e_1}/8$ and local attention length $\log_2(L_{e_1}/8)$ in each subsequence for our sparse attention model and $L_{e_2} = 293$ in the full attention counterpart. For `traffic-f` dataset, we select $L_{t_1} = 576$ with subsequence length $L_{t_1}/8$ and local attention length $\log_2(L_{t_1}/8)$ in each subsequence for our sparse attention model, and $L_{t_2} = 254$ in the full attention counterpart. The calculation of memory usage and other details can be found in Appendix A.4. We conduct experiments on aforementioned sparse and full attention models with/without convolutional self-attention on both datasets. By following such settings, we summarize our results in Table 3 (Upper part). No matter equipped with convolutional self-attention or not, our sparse attention models achieve comparable results on `electricity-f` but much better results on `traffic-f` compared to its full attention counterparts. Such performance gain on `traffic-f` could be the result of the dateset's stronger long-term dependencies and our sparse model's better capability of capturing these dependencies, which, under the same memory budget, the full attention model cannot match. In addition, both sparse and full attention models benefit from convolutional self-attention on challenging `traffic-f`, proving its effectiveness.

To explore how well our sparse attention model performs compared to full attention model with the same input length, we set $L_{e_2} = L_{e_1} = 768$ and $L_{t_2} = L_{t_1} = 576$ on `electricity-f` and `traffic-f`, respectively. The results of their comparisons are summarized in Table 3 (Lower part). As one expects, full attention Transformers can outperform our sparse attention counterparts no matter they are equipped with convolutional self-attention or not in most cases. However, on `traffic-f` dataset with strong long-term dependencies, our sparse Transformer with convolutional self-attention can get better results than the canonical one and, more interestingly, even slightly outperform its full attention counterpart in term of $R_{0.5}$, meaning that our sparse model with convolutional self-attention can capture long-term dependencies fairly well. In addition, full attention models under length constraint consistently obtain gains from convolutional self-attention on both `electricity-f` and `traffic-f` datasets, showing its effectiveness again.

Table 3: Average $R_{0.5}/R_{0.9}$-loss comparisons between sparse attention and full attention models with/without convolutional self-attention by rolling-day prediction of 7 days. "Full" means models are trained with full attention while "Sparse" means they are trained with our sparse attention strategy. "+ Conv" means models are equipped with convolutional self-attention with kernel size $k = 6$.

| Constraint | Dataset | Full | Sparse | Full + Conv | Sparse + Conv |
|---|---|---|---|---|---|
| Memory | electricity-f$_{1d}$ | 0.083/0.051 | 0.084/**0.047** | **0.078**/0.048 | 0.079/0.049 |
| | traffic-f$_{1d}$ | 0.161/0.109 | 0.150/0.098 | 0.149/0.102 | **0.138/0.092** |
| Length | electricity-f$_{1d}$ | 0.082/0.047 | 0.084/0.047 | **0.074/0.042** | 0.079/0.049 |
| | traffic-f$_{1d}$ | 0.147/0.096 | 0.150/0.098 | 0.139/**0.090** | **0.138**/0.092 |

**Further Exploration** In our last experiment, we evaluate how our methods perform on datasets with various granularities compared to our baselines. All datasets except `M4-Hourly` are evaluated by rolling window 7 times since the test set of `M4-Hourly` has been provided. The results are shown in Table 4. These results further show that our method achieves the best performance overall.

Table 4: $R_{0.5}/R_{0.9}$-loss of datasets with various granularities. The subscript of each dataset presents the forecasting horizon (days). TRMF is not applicable for `M4-Hourly`$_{2d}$ and we leave it blank. For other datasets, TRMF outputs points predictions, so we only report $R_{0.5}$. $^\diamond$ denotes results from [10].

|        | electricity-f$_{1d}$ | traffic-f$_{1d}$ | solar$_{1d}$ | M4-Hourly$_{2d}$ | wind$_{30d}$ |
|--------|--------------|-----------|---------|-----------|--------|
| TRMF   | 0.094/-      | 0.213/-   | 0.241/- | -/-       | 0.311/- |
| DeepAR | 0.082/0.063  | 0.230/0.150 | 0.222/0.093 | 0.090$^\diamond$/0.030$^\diamond$ | 0.286/0.116 |
| Ours   | **0.074/0.042** | **0.139/0.090** | **0.210 /0.082** | **0.067 /0.025** | **0.284/0.108** |

## 6 Conclusion

In this paper, we propose to apply Transformer in time series forecasting. Our experiments on both synthetic data and real datasets suggest that Transformer can capture long-term dependencies while LSTM may suffer. We also showed, on real-world datasets, that the proposed convolutional self-attention further improves Transformer' performance and achieves state-of-the-art in different settings in comparison with recent RNN-based methods, a matrix factorization method, as well as classic statistical approaches. In addition, with the same memory budget, our sparse attention models can achieve better results on data with long-term dependencies. Exploring better sparsity strategy in self-attention and extending our method to better fit small datasets are our future research directions.

## Footnotes

[1]Here time index $t$ is relative, i.e. the same $t$ in different time series may represent different actual time point.

[2]Since the model is applicable to all time series, we omit the subscript $i$ for simplicity and clarity.

[3]By referring to Transformer, we only consider the autoregressive Transformer-decoder in the following.

[4]At each time step the same model is applied, so we simplify the formulation with some abuse of notation.

[5]Applying other bases is trivial so we don't discuss other bases here for simplicity and clarity.

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
