[Supplementary Material · NeurIPS2019-12-14.pdf]

# A Supplementary Materials

## A.1 Proof of Theorem 1

*Proof.* According to the attention strategy in *LogSparse Transformer*, in each layer, cell $l$ could attend to the cells with indicies in $I_l^k = \{l - 2^{\lfloor \log_2 l \rfloor}, l - 2^{\lfloor \log_2 l \rfloor - 1}, l - 2^{\lfloor \log_2 l \rfloor - 2}, \cdots, l - 2^0, l\}$. To ensure that every cell receives the information from all its previous cells and itself, the number of stacked layers $\tilde{k}_l$ should satisfy that $S_l^{\tilde{k}_l} = \{j : j \le l\}$ for $l = 1, \cdots, L$. That is, $\forall\, l$ and $j \le l$, there is a directed path $P_{jl} = (j, p_1, p_2, \cdots, l)$ with $\tilde{k}_l$ edges, where $j \in I_{p_1}^1$, $p_1 \in I_{p_2}^2, \cdots, p_{\tilde{k}_l - 1} \in I_l^{\tilde{k}_l}$. We prove the theorem by constructing a path from cell $j$ to cell $l$, with length (number of edges) no larger than $\lfloor \log_2 l \rfloor + 1$. Case $j = l$ is trivial, we only need to consider $j < l$ here. Consider the binary representation of $l - j$, $l - j = \sum_{m=0}^{\lfloor \log_2 (l-j) \rfloor} b_m 2^m$, where $b_m \in \{0, 1\}$. Suppose $\{m_{sub}\}$ is the subsequence $\{m | 0 \le m \le \lfloor \log_2 (l - j) \rfloor, b_m = 1\}$ and $m_p$ is the $p_{th}$ element of $\{m_{sub}\}$. A feasible path from $j$ to $l$ is $P_{jl} = \{j, j + 2^{m_0}, j + 2^{m_0} + 2^{m_1}, \cdots, l\}$. The length of this path is $|\{m_{sub}\}|$, which is no larger than $\lfloor \log_2 (l - j) \rfloor + 1$. So

$$\min \{\tilde{k}_l | S_l^{\tilde{k}_l} = \{j : j \le l\}\} \le \max_{\{j | j < l\}} \lfloor \log_2 (l - j) \rfloor + 1 \le \lfloor \log_2 l \rfloor + 1.$$

Furthermore, by reordering $\{m_{sub}\}$, we can generate multiple different paths from cell $j$ to cell $l$. The number of feasible paths increases at a rate of $O(\lfloor \log_2 (l - j) \rfloor!)$ along with $l$. □

## A.2 Training

Table 5: Dataset statistics. "T", "M" and "S" represent the length, number and sample rate of the time series, respectively.

|   | electricity-c | electricity-f | traffic-c | traffic-f | wind | solar | M4-Hourly |
|---|---|---|---|---|---|---|---|
| T | 32304 | 129120 | 4049 | 12435 | 10957 | 5832 | 748/1008 |
| M | 370 | 370 | 963 | 963 | 28 | 137 | 414 |
| S | 1 hour | 15 mins | 1 hour | 20 mins | 1 day | 1 hour | 1 hour |

To learn the model, we are given a time series dataset $\{\mathbf{z}_{i,1:T}\}_{i=1}^M$ and its associated covariates $\{\mathbf{x}_{i,1:T}\}_{i=1}^M$, where $T$ is the length of all available observations and $M$ is the number of different time series. The dataset statistics is shown as Table 5. Following [3], we create training instances by selecting windows with fixed history length $t_0$ and forecasting horizon $\tau$ but varying the start point of forecasting from each of the original long time series. As a follow-up of [3], we sample training windows through weight sampling strategy in [3]. Note that during selecting training windows, data in the test set can never be accessed. As a result, we get a training dataset with $N$ sliding windows $\{\mathbf{z}_{i,1:t_0+\tau}, \mathbf{x}_{i,1:t_0+\tau}\}_{i=1}^N$.

For positional encoding in Transformer, we use learnable *position embedding*. For covariates, following [3], we use all or part of *year*, *month*, *day-of-the-week*, *hour-of-the-day*, *minute-of-the-hour*, *age* and *time-series-ID* according to the granularities of datasets [3]. *age* is the distance to the first observation in that time series [3]. Each of them except *time series ID* has only one dimension and is normalized to have zero mean and unit variance (if applicable). For *time-series-ID*, it has the same dimension as *position embedding* through *ID embedding* matrix so that they can be summed up (with broadcasting). The summation is then concatnated with aforementioned other covariates as the input of 1st layer in Transformer.

`DeepAR` [3] uses an encoder-decoder fashion, where the encoder is the same as the decoder and the final hidden state of the encoder is used to initialize the hidden state of the decoder. Such an architecture can be seen as a decoder-only network as the encoder and decoder are the same, where the objective is to predict the distribution of next point according to current input and last hidden state. Inspired by this observation, we use Transformer decoder-only mode [36] to model time series. Similar to [37], a fully-connected layer on the top of Transformer is stacked, which outputs the parameters of the probability distribution after scaling for the next time point with appropriate transformations. For example, for parameters requiring positivity, a softplus activation is applied. We use the same scale handling technique as in [3] to scale our input and output of our models. We refer readers to [3] for more details of scale handling. In our experiments, we use Gaussian likelihood since our training datasets are real-valued data. Note that one can also use other likelihood models, e.g. negative-binomial likelihood for positive count data. In synthetic datasets, we only count log-likelihood from $t_0 + 1$ to $t_0 + \tau$. On real-world datasets, we not only count log-likelihood from $t_0 + 1$ to $t_0 + \tau$, but also include the log-likelihood from $1$ to $t_0$, similar to training in [3] and pre-training in [37].

During training, we use vanilla Adam optimizer [28] with early stopping except experiments on `electricity-f` and `traffic-f` to maximize the log-likelihood of each training instance. Our preliminary study show that

training on these two datasets are very unstable with Adam. Rather, we found that BERTAdam [38] [8], a variant of Adam with warmup and learning rate annealing, can stabilize the training process on these two datasets.

For `electricity-c` and `traffic-c`, we take 500K training windows while for `electricity-f` and `traffic-f`, we select 125K and 200K training windows, respectively. For `wind`, `M4-Hourly` and `solar`, we choose 10K, 50K and 50K training windows, respectively. The window selection strategy is described above. For our Transformer models, all of them use $H = 8$ heads and the dimension of *position embedding* and *time series ID embedding* are all 20. All of our models have 3 layers except experiments on `electricity-f` and `traffic-f`, where our models use 6 and 10 layers, respectively. The data before the forecast start time is used as the training set and split into two partitions. For each experiment on real-world datasets, we train our model on the first partition of the training set containing 90% of the data 5 times with different random seeds and we pick the one that has the minimal negative log-likelihood on the remaining 10%. The results on test set corresponding to minimal negative log-likelihood on the remaining 10% are reported. All models are trained on GTX 1080 Ti GPUs.

## A.3 Evaluation

Following the experimental settings in [6], one week data from 9/1/2014 00:00 (included) [9] on `electricity-c` and 6/15/2008 17:00 (included) [10] on `traffic-c` is left as test sets. For `electricity-f` and `traffic-f` datasets, one week data from 8/31/2014 00:15 (included) and 6/15/2008 17:00 (included) [11] is left as test sets, respectively. For `solar`, we leave the last 7 days in August as test set. For `wind`, last 210 days in year 2015 are left as test set. For `M4-Hourly`, its training and test set are already provided. After training on previous settings, we evaluate our models on aforementioned test intervals and report standard quantile loss ($R_{0.5}$ and $R_{0.9}$) on the test sets.

## A.4 Implementation of sparse attention and its memory cost

During the implementation of our sparse attention, $\forall l \leq |I_L^k|$, one can allow such cell $l$ to densely attend all its past cells and itself without increasing space usage as query-key matching are parallelly computed in reality and maximum number of cells that a cell can attend is reached by cell $L$.

Our current implementation of *LogSparse* attention is via a mask matrix and its relative memory usage is calculated ideally from the attention matrix, which is the memory bottleneck of Transformer.

For `electricity-f` dataset, we choose $L_{e_1} = 768$ with subsequence length $L_{sub}^{e_1} = L_{e_1}/8 = 96$ and local attention length $L_{loc}^{e_1} = \lceil \log_2(L_{sub}^{e_1}) \rceil = 7$ in each subsequence for our sparse attention model, and $L_{e_2} = 293$ in its full attention counterpart. We stack the same layers on both sparse attention and full attention models. Hence, we can make sure that their whole memory usage is comparable if their memory usage is comparable in every layer. In sparse attention equipped with local attention, every cell attends to $2 * L_{loc}^{e_1} = 14$ cells in each subsequence at most, causing a cell attend to $2 * L_{loc}^{e_1} * L_{e_1}/L_{sub}^{e_1} = 14*8 = 112$ cells at most in total. Therefore, we get the memory usage of sparse attention in each layer is $L_{e_1} * 2 * L_{loc}^{e_1} * L_{e_1}/L_{sub}^{e_1} = 768*112 = L_{e_2}^2 \approx 293$. Following such setting, the memory usage of the sparse attention model is comparable to that of the full attention model. For `traffic-f` dataset, one can follow the same procedure to check the memory usage.

## A.5 Visualization of attention matrix

Here we show an example of learned attention patterns in the masked attention matrix of a head within canonical Transformer's last layer on `traffic-c` dataset. Figure 6 (a) is a time series window containing 8 days in `traffic-c`. The time series obviously demonstrates both hourly and daily patterns. From its corresponding masked attention matrix, as shown in Figure 6 (b), we can see that for points in weekdays, they heavily attend to previous cells (including itself) at the same time in weekdays while points on weekends tend to only attend to previous cells (including itself) at the same time on weekends. Hence, the model automatically learned both hourly and daily seasonality, which is the key to accurate forecasting.

Figure 6: (a): An example time series window in `traffic-c` dataset. (b): Corresponding learned attention patterns in the masked attention matrix of a head within the last layer.

## Footnotes

[8] `https://github.com/nlpdata/mrc_bert_baseline/blob/master/bert/optimization.py`

[9] Value in 00:00 is the aggregation of original value in 00:15, 00:30, 00:45 and 01:00.

[10] Value in 17:00 is the mean of original value in 17:00, 17:10, 17:20, 17:30, 17:40 and 17:50.

[11] Value in 17:00 is the mean of original value in 17:00 and 17:10.