[Reviews · NeurIPS 2019]

Reviewer 1



This paper proposes to use causal convolutions to enhance local structure and a log-sparse attention formulation to reduce the memory requirements of Transformers. Moreover, different log-sparse variations are proposed, for example local and restart. The experiments involve exploring different convolutional kernel sizes, the combination of convs and log-sparse, and evaluating the data at different resolutions. Pros: - Addresses key challenges in Transformers, enhancing locality and reducing memory through log-sparse formulation. - The log-sparse formulation is intuitive as dense on recent history and sparser as history is more distant. - The figure illustrations such as Figure 1, 2, and 3 are extremely informative. Cons: - The variations in the log-sparse formulation such as local and restart does not seem to be tested in the experiments. - Perhaps provide some plots for the real data against generated sequences to help readers see the challenges in the dataset (such as how frequently do changes points happen) and qualitatively how well the model is able to capture them.

Reviewer 2



On its own, the paper assigns text space inefficiently. While simulated data is fine (in context) it should not take space from essential Methods description. The baseline method is described in one short and incomplete paragraph, insufficiently described apart from a couple of perfunctory references (one of which, as several other references, is incompletely listed). The architecture window does not suffice and it is left to the reader to piece together how the forecasts are computed from data, end-to-end. Several mentions of 'rolling window' is not a sufficient description of train/validation/test procedure. What was it? Depending on the exact details, the evaluation procedure can result in overfit. What was the loss function used in training (it appears only briefly in the 'training curve' Figure). Results are incomplete. For the M4 dataset in particular (which has a test set), there are known accuracy results in the literature which can be compared with the R_0.5 result. They should be included. There are scant or no details given on how alternative methods (Arima, TRMF, DeepAR) have been set up (lag length, metaparameters) or how the metaparemeters of the proposed method (in particular the kernel size) has been chosen *prior* to any ablation studies Finally, while the stated goal is computational efficiency, not running time is reported, nor the actual software/hdwre architecture that implemented the main method. Minor details, for improved clarity: the methods section is minuscule: less than 10 lines on Page 3. Expand Deep neural networks have been proposed to capture shared information across related time series for accurate forecasting. how were baseline metaparameters (kernel size, h) chosen before ablation study? Figure 5 suggests NN was trained iteratively over the same data ARIMA performs significantly worse than the simpler method (ETS) which suggests seasonality was not used in what software / environment was main method and improvement implement and how fast did these run?

Reviewer 3



The paper is a rather straightforward extension of the well known transformer network for time series forecasting. However, it precisely targets two major limitations of the original algorithm, and the proposed improvements handle them effectively. As a result, it shows a significant improvement over the state-of-the-art methods such as DeepAR, especially for datasets which require long term dependency modeling. The paper is clearly written and the quality is high in all aspects. Readers can understand the benefit of each proposed component, thanks to carefully designed experiments. I think it is a significant contribution for the community, demonstrating the potential of transformer networks for time series forecasting. Some questions: Could you provide the dimension of covariate vectors x for each of the experiments with some details? Which positional encoding scheme was used? Exactly the same as the formula in p6 of the original transformer paper? It seems the performance of the proposed algorithm for electricity-f_1d and traffic-f_1d in Table 3 and 4 do not match. Is the kernel size different? What was the window size for electricity-c and traffic-c experiments? The full history length? Could you provide more details? Would the performance with k = 1 in Table 2 be almost the same as the one of the original transformer network? I assume a sparse attention was used in Table 2, but the performance should be equivalent or better with full attention model according to Table 3. Do you have exact numbers for the original transformer network?

[Author Response · NeurIPS 2019]

**To R #1 and R #2** for dual submission concerns: Although both papers are among the first to explore Transformer in time series forecasting and achieve SOTA results, these two papers investigate different problems of time series with Transformer (long-term dependencies & memory V.S. abrupt changes). ❶ As mentioned by **R #3**, this paper conducts experiments on both synthetic and real-world datasets to demonstrate the superior of Transformer to LSTM on forecasting time series with long-term dependencies, which submission 6113 (S6113) does not mention. ❷ This paper pays much attention to memory issues in Transformer and develops *LogSparse* Transformer to break its notorious memory bottleneck from $O(L^2)$ to $O(L(\log_2 L)^2)$ with theoretical justification, which S6113 also does not deal with. ❸ S6113 focuses on how to enable fast responses to abrupt changes in time series, which this paper does not discuss.

As for *causal convolution*, both papers have their own motivations (robustness V.S. quick response) to use it and utilize different solutions to achieve their corresponding goals. ❶ `Motivation`: This paper uses it to generate more local context-aware queries and keys in **EVERY** layer so that they can be matched by referring to local information, enhancing the robustness of Transformer. In contrast, S6113 deploys it only in **First** layer to equip the model with capability to adapt to abrupt changes quickly. ❷ `Solutions`: In this paper, queries and keys are produced by $Q = \mathbf{Conv}_k(Y)$ and $K = \mathbf{Conv}_k(Y)$ in every layer to enhance the robustness of Transformer while S6113 only extracts features from **RAW DATA** for abrupt changes by $F = \mathbf{Conv}_k(Z) + \mathbf{Conv}_1(Z)$ in the first layer and then produces them by $Q = W_K F$ and $K = W_V F$ as in the standard Transformer.

**To R #1**: ❶ `Empirical contribution & Bridging fields`: Time series forecasting has been extensively studied in the past few decades and RNNs have been the new norms in modern large scale forecasting tasks [3] and [6]. However, we figure out their long-term modeling issues through carefully designed experiments and only use simple modified Transformer networks to outperform existing works. As elaborated by **R #3**, these results bring fresh air to time series forecasting and demonstrate the great potential of Transformer, which is our main contribution. ❷ `Dataset overlap`: The only overlapping datasets with S6113 are `electricity-c` and `traffic-c`. They are publicly available and extensively used in [3], [6], [7], [9] and [17]. In addition, for comparison with [6], whose source code is not available after contacting its authors, we need to run experiments on them. Moreover, this paper also includes experiments on five other non-overlapping datasets. ❸ `Local and Restart attention`: They are already used in our experiments as elaborated in Sec. 5.2 *Sparse attention*. We plan to add ablation study to illustrate this in the new version. ❹ `Sample plot`: We will add some plots to help readers see the challenges and how well the model can capture them.

**To R #2**: We are sorry that we have to simplify many details due to space limitations and place them in the Appendix. ❶ *Text space:* We demonstrate simulated data in details since it is used to quantitatively prove the superior of Transformer to LSTM in capturing long-term dependencies, a key component of our story. We will try to find a better way to present this part clearly and concisely. ❷ *Rolling window:* It is used in [3], [6] and [17], our main baselines. Taking rolling-day predication of **7** days as an example, prediction horizon is one day and forecasts start time is shifted by one day **7** times after evaluating the prediction for the current day [6]. ❸ Loss function, train/validation/test split procedure, hardware, training procedures, optimizer, evaluation, hyperparameters, and other details are elaborated in Appendix A.2 and A.3. Note that loss function, split procedure and some other details are the same as [3] for fair comparisons. Test set (interval) is chosen by [3] and [6], which won't be preferable to our model, and will never be accessed by the model during training. ❹ For *TRMF*, we change `lag_set` according to the periods of our data as indicated in [17] and note that the results we report are much better than those in [3] and [6]. For *DeepAR*, we use results in [6] if it reports, otherwise fine tune its hidden size and learning rate, and report the best performance according to the validation set. For *kernel size*, we try {1,2,3,6,9} and report their best performance according to validation set. Thanks to the potential of Transformer, even though we almost didn't tune any hyperparameters, e.g. learning rate, layers, and hidden size, it can still achieve better results than SOTA.

**To R #3**: ❶ `Positional encoding`: We use learnable `position embedding`. ❷ `Covariate`: Following [3], we use all or part of `year`, `month`, `day_of_week`, `hour`, `minute`, `absolute_position` and `time_series_ID` according to the granularities of datasets. Each of them except `time_series_ID` has only one dimension and is normalized to have zero mean and unit variance (if applicable). For `time_series_ID`, it has the same dimension as `position embedding` through `ID_embedding` matrix so that they can be summed up (with broadcasting). The summation is then concatnated with aforementioned other covariates as the input of 1st layer in Transformer. We will add a table in Appendix to elaborate more details for each dataset in the new version. ❸ `Mismatch`: Yes, they used different kernel sizes and we will clarify this in the new version. ❹ `Window size`: For `electricity-c` and `traffic-c`, their window sizes are $192 = 168$ (one week) $+ 24$ (one day). The full history is split into short windows for efficient training (for example, directly feeding hourly data with one year history into LSTM or Transformer is impossible). This is a follow-up of [3]. The window selection procedures and other details are elaborated in Appendix A.2. During evaluation, we feed the last one week data before test interval into the model and do prediction. ❺ `Questions about Table 2`: Its results are from models trained with full attention as described in Sec. 5.2 *Convolutional self attention*. Therefore, with $k = 1$, it is the original Transformer. ❻ `Source code`: We plan to release our source code if it is accepted to help the community testify ideas quickly.

[Meta-Review · NeurIPS 2019]

Since this paper is similar to paper ID 6113, this weakens the novelty of the paper as set out in the dual submission policy. However, the paper has more novel contributions over 6113, such as using k convolutions instead of 1 convolution. The paper would also benefit from a more complete description of the main method.